# Biotechnological Fungal Platforms for the Production of Biosynthetic Cannabinoids

**DOI:** 10.3390/jof9020234

**Published:** 2023-02-10

**Authors:** Katarina Kosalková, Carlos Barreiro, Isabel-Clara Sánchez-Orejas, Laura Cueto, Carlos García-Estrada

**Affiliations:** 1INBIOTEC (Instituto de Biotecnología de León), Av. Real 1, 24006 León, Spain; 2Área de Bioquímica y Biología Molecular, Departamento de Biología Molecular, Campus de Vegazana, Universidad de León, 24007 León, Spain; 3Departamento de Ciencias Biomédicas, Campus de Vegazana, Universidad de León, 24007 León, Spain

**Keywords:** cannabinoids, fungal platforms, genetic engineering, *Penicillium chrysogenum*, yeasts

## Abstract

Cannabinoids are bioactive meroterpenoids comprising prenylated polyketide molecules that can modulate a wide range of physiological processes. Cannabinoids have been shown to possess various medical/therapeutic effects, such as anti-convulsive, anti-anxiety, anti-psychotic, antinausea, and anti-microbial properties. The increasing interest in their beneficial effects and application as clinically useful drugs has promoted the development of heterologous biosynthetic platforms for the industrial production of these compounds. This approach can help circumvent the drawbacks associated with extraction from naturally occurring plants or chemical synthesis. In this review, we provide an overview of the fungal platforms developed by genetic engineering for the biosynthetic production of cannabinoids. Different yeast species, such as *Komagataella phaffii* (formerly *P. pastoris*) and *Saccharomyces cerevisiae*, have been genetically modified to include the cannabinoid biosynthetic pathway and to improve metabolic fluxes in order to increase cannabinoid titers. In addition, we engineered the filamentous fungus *Penicillium chrysogenum* for the first time as a host microorganism for the production of Δ^9^-tetrahydrocannabinolic acid from intermediates (cannabigerolic acid and olivetolic acid), thereby showing the potential of filamentous fungi as alternative platforms for cannabinoid biosynthesis upon optimization.

## 1. Introduction

*Cannabis sativa* produces about 100 (alkyl-)phytocannabinoids (meroterpenoids and their analogs derived from cannabis), generically known as cannabinoids [1,2,3]. Cannabis is supposedly of Asian origin and is one of humanity’s oldest cultivated crops [4,5,6]. It is an annual, usually dioecious and sporadically monoecious, wind-pollinated species that is extremely allogamous (cross-fertilized) in nature [2]. The taxonomy of the *Cannabis* genus is complex due to the existence of cultivated and wild varieties, as well as its economic importance and uses (hemp, recreational or medical). The current botanical nomenclature [7] includes just one species in the genus, *C. sativa*, and two subspecies (subsp. *sativa* (fibers and oil cultivars) and subsp. *indica* (“narcotic” cultivars)) based on the THC (tetrahydrocannabinol) content (hemp varieties should not contain more than 0.3% THC in dried female flowering tops). The domestication status is also used as a characteristic to distinguish the two varieties within each subspecies [4,8,9].

The fiber varieties (“hemp”: fiber extracted from the stems of *C. sativa*) present a high carbon-sequestering potential due to their rapid growth, and so can be used as building materials or biofuel. However, the traditional use of fiber includes the manufacture of rope, shoes, canvas, paper, clothing, and sails [4,9,10]. On the other hand, the cannabinoids present in the sticky resin produced by the female plant, including psychoactive compounds such as THC and (−)-trans-Δ^9^-tetrahydrocannabinol (Δ^9^-THC) and non-psychoactive components with potential therapeutic effects (e.g., reducing inflammation, chronic pain, and nausea) such as cannabidiol (CBD) [11,12,13], are other relevant products of *C. sativa*.

## 2. Chemical Structure of Cannabinoids, Types, and Metabolic Routes for Biosynthesis

Phytocannabinoids are bioactive natural meroterpenoids with a resorcinol core bearing a para-positioned isoprenyl, alkyl, or aralkyl side chain [14]. The side chain usually contains an odd number of carbon atoms (cannabinoids containing an even number of carbon atoms are rare). Phytocannabinoids are found in angiosperms, fungi, and liverworts and are produced in several plant organs, such as the flower and glandular trichome of *C. sativa*, the scales of *Rhododendron*, and the oil bodies of *Radula* species. Moreover, it has been found that the mammalian brain has receptors that respond to compounds found in *C. sativa*. These cannabinoid receptors form the endocannabinoid system and regulate several biological functions and metabolic processes. Therefore, these bioactive compounds can be beneficial for the treatment of pain, anxiety, and cachexia in humans [15,16].

The different cannabinoids produced by *C. sativa* [17] are classified into structural families [18,19], such as cannabidiols (CBDs), cannabigerols (CBGs), cannabicyclols (CBLs), cannabinodiols (CBNDs), cannabinols (CBNs), cannabitriols (CBTs), cannabichromenes (CBCs), Δ^9^-THC, and miscellaneous cannabinoids [20]. The acid forms are the final product of the cannabinoid biosynthetic pathways, although several spontaneous modifications such as oxidation, decarboxylation, and cyclization frequently take place because of the poor oxidative stability of phytocannabinoids, in particular Δ^9^-THC [14].

Phytocannabinoids are accumulated in glandular thricomes all over the aerial parts of the plant, especially in the female flowers [21,22]. The phytocannabinoid biosynthetic pathway is split between different cellular compartments and organelles: the cytosol of gland cells (polyketide pathway), the plastids (methylerythritol 4-phosphate (MEP) pathway for prenylation), and the extracellular storage cavity (oxidocyclization and storage) (Figure 1). It is still unclear how intermediates and precursors are transported between compartments, although it is most likely that vesicle trafficking and transport proteins are involved in the movements of these intermediates across the interface between the gland cells and the storage cavity.

Inside the cytosol, the biosynthesis of cannabinoids involves the integration of several steps in polyketide and isoprenoid metabolism. C18 fatty acids are desaturated, preoxygenated, and cleaved into hexanoic acid, which is transformed into thioester hexanoyl-CoA in a reaction catalyzed by acyl-activated enzyme 1 (AAE1). Later, the hexanoyl-CoA is elongated with malonyl-CoA in a reaction catalyzed by olivetol synthase (OLS) and cyclized by olivetolic acid cyclase (OAC) to produce olivetolic acid (OA) [23,24,25].

Inside plastids, geranyl pyrophosphate (GPP) is synthesized through the MEP pathway. OA is prenylated using GPP by cannabigerolic acid synthase (CBGAS), thereby forming cannabigerolic acid (CBGA), the first cannabinoid. CBGA is an essential cannabinoid because it is the precursor of several other cannabinoids. Then, the CBGA is converted into Δ^9^-tetrahydrocannabinolic acid (Δ^9^-THCA) and cannabidiolic acid (CBDA) in the apoplastic space by the action of two enzymes, Δ^9^-THCA synthase (THCAS) and CBDA synthase (CBDAS), respectively. This conversion causes the reduction of O_2_ into hydrogen peroxide (H_2_O_2_) via oxidative cyclization reactions. Another important enzyme, cannabichromenic acid synthase (CBCAS), participates in the synthesis of cannabichromenic acid (CBCA) from CBGA using FAD and O_2_. THCAS, CBGAS, and CBDAS are also flavoproteins, which are strictly dependent on the presence of O_2_ as an electron acceptor. All these oxidocyclases carry a secretion signal peptide and are exported to the extracellular resin space. THCAS and CBDAS are active in the resin space, but it is still unknown whether their activity is exclusive to the extracellular space [23,26,27].

Δ^9^-THCA, CBDA, and CBDA with a pentyl side chain are the end-products of the enzymatic biosynthesis of cannabinoids and are synthesized in the apoplastic space. These active compounds undergo spontaneous rearrangement reactions when exposed to heat or radiation or during storage [23,26,28].

## 3. Importance, Applications, and Impact of Cannabinoids

Phytocannabinoids play several roles in human health, and cannabis preparations have been used in medicines since ancient times [29]. On the one hand, they exhibit anti-microbial activity against some bacteria and fungi, and they are effective against a wide range of infectious diseases in humans, acting as potent antibiotics. Cannabis extracts possess antimicrobial activity against some Gram-positive bacteria, such as *Bacillus subtilis* and *Staphylococcus aureus*, and the Gram-negative bacteria *Escherichia coli* and *Pseudomonas aeruginosa*. In contrast, no activity is displayed against *Candida albicans* and *Aspergillus niger* [30]. Phytocannabinoids (e.g., Δ^9^-THC, CBG, CBN, CBD, and CBC) display antibiotic activity against methicillin-resistant *Staphylococcus aureus* [31]. Δ^9^-THC and CBD exhibit bactericidal activity against streptococci and staphylococci but not against Gram-negative bacteria [32]. Because of their pharmacological potential, an increasing number of countries are relaxing their legislation around phytocannabinoids, and the global industry around cannabis-derived products is growing exponentially [33].

On the other hand, phytocannabinoids exert strong therapeutic potential in humans owing to their interaction with the G-protein-coupled cannabinoid receptors (GPCRs), such as CB1 and CB2; transient receptor potential (TRP) ion channels; and peroxisome proliferator-activated receptor (PPAR). CB1 is the most abundant GPCR in the central nervous system, and CB2 is located predominantly in the cells and tissues of the immune system [34]. Thus, cannabinoids play a key role in signaling, and the proper functioning of the immune and central nervous systems. Δ^9^-THC is the major psychoactive component of cannabis and displays pleiotropic effects in humans, including analgesic response, relaxation, pain tolerance, and dysphoria (anxiety disorder). Δ^9^-THC is also used by patients with insomnia and depression, since it improves sleep [35]. CBD exerts its function in humans through the CB1 and CB2 receptors in the CNS and the peripherical regions [36]. It is administered to patients with treatment-resistant epilepsy. CBD can also be delivered to patients receiving pharmacotherapy and can act as a potential cannabinoid to cure obesity, convulsive disorder, and rheumatoid arthritis. Furthermore, CBD also exhibits anti-psychotic, anti-nausea, and anti-anxiety properties [37].

In addition to the legal cannabis market (including medical and recreational uses), the EMCDDA (European Monitoring Centre for Drugs and Drug Addiction) estimated the illicit cannabis market in 2017 to be around 1.4–1.7 tones, valued at EUR 10.5–12.8 billion [38]. Several clinical uses of cannabinoids have been described, but just a few are legally recognized by international regulatory agencies. The possible clinical applications of CBD have been summarized in sixteen diseases, which included several cancer types through anti-proliferative and anti-invasive action; inflammatory bowel and Chron’s diseases; cardiovascular diseases through anti-oxidant and anti-inflammatory properties; and neurodegenerative diseases (e.g., Parkinson’s and Alzheimer’s) [13]. However, in January 2020 the US Food and Drug Administration (FDA), FDA and Cannabis: Research and Drug Approval Process: https://www.fda.gov/news-events/public-health-focus/fda-and-cannabis-research-and-drug-approval-process (accessed on 1 February 2023) indicated that up to that date just one cannabis-derived drug product had been approved, Epidiolex, containing a purified form of CBD involved in the treatment of seizures associated with Lennox–Gastaut syndrome or Dravet syndrome in patients 2 years of age and older. In addition, three synthetic cannabis-related drug products have been validated by the FDA: (i) Marinol (dronabinol); (ii) Syndros (dronabinol); and (iii) Cesamet (nabilone). On the one hand, Marinol and Syndros represent a synthetic version of THC, the psychoactive component of cannabis, for the treatment of nausea due to cancer chemotherapy and anorexia, thus preventing weight loss in AIDS patients. On the other hand, Cesamet has a chemical synthetic structure similar to THC as an active ingredient, which is used to treat nausea associated with cancer chemotherapy.

Thus, in summary, cannabis as a hemp fiber is an economical reality, but the most promising market comprises the medical and recreative uses of the plant. However, the botanical derivatives are scarce in the legal market due to the putative risk of co-purification with other cannabinoids during the extraction process from naturally occurring plants. Nowadays, synthetic products are on the market through companies such as AbbVie Inc.; Corbus Pharmaceuticals Holdings Inc.; INSYS Therapeutics Inc.; and Bausch Health, although chemical synthesis is a costly process involving the use of chemicals that are not environmentally friendly. Nevertheless, synthetic production has paved the way for the biotechnological production of single cannabinoid compounds or derivatives in microorganisms, mainly fungi, which have been endowed with the biochemical machinery to reproduce the plant pathways and products [39], an approach that has been promoted by the development of recent synthetic biology methodologies [40].

## 4. Development of Biotechnological Fungal Platforms for the Biosynthesis of Cannabinoids

Fungi produce a broad diversity of primary and secondary metabolites with different bioactivities. Different biotechnological processes have been implemented with the yeasts *Saccharomyces cerevisiae* [41,42,43], *Yarrowia lipolitica* [44,45], and *Komagataella phaffii* (formerly *Pichia pastoris* [46,47]) [48,49]. The production of high-value metabolites through different biotechnological processes is also achieved by means of filamentous fungi, such as *Aspergilli* and *Penicillium* species, *Acremonium chrysogenum*, *Blakeslea trispora,* and *Trichoderma reesei* [50,51,52]. Among these filamentous fungi, *P. chrysogenum* (*P. rubens*) stands out for its adaptability to the massive industrial overproduction of penicillin [53,54,55], which suggests the use of this microorganism as a cell factory for products other than β-lactam antibiotics [56,57,58]. The complete genome sequences of yeasts and filamentous fungi obtained in recent years have provided the basic information for the development of new approaches aimed at improving production strains, increasing metabolite yields, and designing new fungal production platforms for natural products with potential commercial value [51,59,60].

Although bacteria such as *Escherichia coli*, *Bacillus subtilis*, *Streptomyces lividans*, and *Corynebacterium glutamicum* are easier to manipulate genetically, these microorganisms are not always suitable for the heterologous expression of proteins of eukaryotic origin due to their lack of post-translational modifications and their limited ability to secrete heterologous proteins. On the contrary, some yeasts and filamentous fungi have the ability to biosynthesize biologically active proteins with a degree of glycosylation similar to the original proteins [52] and possess a great secreting capacity for different enzymes and metabolites. It should be noted that 82% of the commercial enzymes used in the food industry are manufactured using fungal hosts [61], which confirms the role of these microorganisms as cell factories for heterologous protein production [62].

The application of new genetic engineering techniques at the molecular level has allowed a considerable increase in the production of recombinant proteins and secondary metabolites in yeast and filamentous fungi [49,63,64,65,66,67]. This has contributed to the design of fungal hosts for the heterologous expression of biosynthetic gene clusters that could work as efficient platforms to produce high-added-value secondary metabolites, such as cannabinoids.

### 4.1. Biosynthesis of Cannabinoids in Yeasts

Cannabinoid biosynthesis by means of biotechnological approaches and independently of the plant has been boosted by: (i) its interest as pharmaceutical and bioactive compounds; (ii) the limiting production of synthetic THC and; (iii) the legal regulatory issues for the cultivation of *C. sativa* in most countries. Thus, numerous papers and patents describing biotechnological procedures for the microbial production of cannabinoids (mainly in yeasts) have been reported in the recent years, as it is summarized below.

The identification of the cannabinoids biosynthetic genes and enzymes [24,68,69,70,71,72] paved the way to reconstruct the biosynthetic pathway and engineer the metabolism in suitable heterologous systems [39], mainly yeasts (Figure 2).

Early reports focused on the recombinant expression of THCAS in different hosts, such as tobacco hairy roots, insect cell cultures, and *K. phaffii* cultures [71,73]. The expression of THCAS in these hosts was very low, although it opened the door to the possibility of biotechnological Δ^9^-THCA production by means of biotechnological platforms. *E. coli* was initially used as a potential host microorganism, but it was quickly discarded, since no functional expression of THCAS was achieved in this microorganism [74,75], thereby suggesting that a eukaryotic expression system was necessary for appropriate enzyme folding and maturation. Initial drawbacks of the THCAS scale-up processes regarding the water solubility of CBGA and the membrane localization of CBGAS led scientists to also focus on the intracellular expression of THCAS in *S. cerevisiae* and *K. phaffii* cells and the possible application of a whole-cell production system for Δ^9^-THCA [75]. These authors used the signal peptide of the vacuolar proteinase A to target THCAS in the cell vacuole and found that the highest enzyme activity was obtained in cultures from proteinase A knockout strains of *K. phaffii.* The further optimization of THCAS expression in *K. phaffii* dramatically increased enzyme activity in relation to previous studies, and the bioconversion of CBGA in whole yeast cells led to the production of 1 mM Δ^9^-THCA [75].

The CBGAS expression problems related to its membrane localization led scientists to use the soluble aromatic prenyltransferase from *Streptomyces* sp. CL190 strain (StNphB) instead of plant prenyltransferase. The partial reconstruction of the Δ^9^-THCA biosynthetic pathway in *K. phaffii* gave rise to the successful synthesis of this compound from OA and GPP. Although the side-product 2-O-geranyl OA was also produced because of StNPhB activity, these results suggested that NphB was a promising candidate for the substitution of plant-derived membrane-bound prenyltransferases in the heterologous system [76].

Although *K. phaffii* was found to be a good host for the heterologous expression of cannabinoid biosynthetic enzymes, several bottlenecks attributed to THCAS protein folding were also observed. They were counteracted by the co-production of different proteins, including foldases and chaperones. Thus, the highest THCAS activity was obtained with the co-production of the spliced gene version of: (i) basic leucine zipper transcription factor Hac1s (involved in the regulation of several genes involved in protein folding, secretion, ER quality control, glycosylation, and ER-associated degradation); (ii) CNE1 (an ER membrane protein that functions as a constituent of the ER quality control apparatus); and (iii) FAD1 (FAD synthetase, involved in the production of FAD from FMN). This strain showed improved productivity in comparison to previous strains and reached 3.05 g/L THCA from the bioconversion of CBGA within 8 h of incubation [77]. The same authors also carried out site-directed mutagenesis in THCAS and CBDAS in order to elucidate the structure–function relationship and improve the enzyme properties for biotechnological cannabinoid production. After the recombinant expression of different mutant enzymes in *K. phaffi*, the enzyme bearing the double substitution N89Q + N499Q (lacking two glycosylation sites) gave rise to a 2.0-fold increase in THCAS activity, whereas the variant A414V (mutant in the active site) was able to increase the catalytic activity for the production of CBDA by 3.3 times [78].

In parallel, *S. cerevisiae* was engineered to generate strains capable of performing the complete biosynthesis of CBGA, Δ^9^-THCA, CBDA, Δ^9^-tetrahydrocannabivarinic acid, and cannabidivarinic acid from the simple sugar galactose [23]. In this biosynthetic platform, hexanoyl-CoA was produced either by feeding hexanoic acid as a substrate for the acyl-activating enzyme (encoded by CsAAE1 from *C. sativa*) or from acetyl-CoA (derived from sugars) due to the introduction of a multiorganism heterologous biosynthetic pathway comprising RebktB from *Ralstonia eutropha*, CnpaaH1 from *Cupriavidus necator*, Cacrt from *Clostridium acetobutylicum*, and Tdter from *Treponema denticola*. In addition to this, the native mevalonate pathway was modified, including a mutant version of Erg20 (F69W/N127W) to overproduce GPP. The introduction of the OA biosynthetic genes CsTKS and CsOAC, a gene encoding a previously undiscovered enzyme with geranylpyrophosphate:olivetolate geranyltransferase activity (CsPT4) involved in CBGA biosynthesis, together with THCAS and CBDAS (with a vacuolar localization tag), allowed *S. cerevisiae* to produce 1.1 mg/L Δ^9^-THCA or 4.3 μg/L CBDA from hexanoic acid, or 2.3 mg/L Δ^9^-THCA or 4.2 μg/L CBDA from galactose. These strains also produced Δ^9^-tetrahydrocannabivarinic acid and cannabidivarinic acid. The titers of Δ^9^-THCA and Δ^9^-tetrahydrocannabivarinic acid were further increased up to 8.0 mg/L (Δ^9^-THCA) and 4.8 mg/L (Δ^9^-tetrahydrocannabivarinic acid) after the overexpression of additional single copies of the biosynthetic genes CsTKS, CsOAC, and THCAS. Moreover, unnatural cannabinoid analogs with modifications in the part of the molecule that is known to alter receptor binding affinity and potency were obtained by feeding different fatty acids (e.g., pentanoic acid, heptanoic acid, and 5-hexenoic acid) to the *S. cerevisiae* engineered strains [23].

The reconstruction of the complete Δ^9^-THCA and CBDA biosynthetic pathway in baker’s yeast represented a milestone in the field of the heterologous biosynthesis of cannabinoids, but the titers were considered low from an industrial point of view. Therefore, bioengineering studies and the modeling of the heterologous biosynthesis of Δ^9^-THCA were carried out in this yeast chassis. Several critical aspects that require systematic optimization were detected after the analysis of metabolic bottlenecks, such as: (i) insufficient hexanoic acid formation in the fatty acid biosynthesis; (ii) low acetyl-CoA precursor delivery to the hexanoic acid biosynthesis and the mevalonate pathway; (iii) the limited catalytic activity of NphB or CsPT and THCAS; (iv) insufficient ATP and NADPH regeneration and; (v) ethanol production by Crabtree effect [79]. In another study, *S. cerevisiae* was engineered by introducing plant uridine diphosphate glycosyltransferases into its genome in order to produce OA glucoside and cannabigerolic acid glucoside. This could be of interest due to the high hydrophobicity of naturally occurring cannabinoids, which affects their use as pharmaceuticals [80].

Due to the importance of yeast peroxisomes in different metabolic pathways, such as the β-oxidation of fatty acids leading to the creation of a pool of acetyl-CoA that can be used in the formation of derivatives [81] (e.g., isoprenoid via the mevalonate pathway), these organelles were engineered for the more efficient production of CBGA. Thus, GPP synthase Erg20p (N127W) and geranyldiphosphate:olivetolate geranyltransferase CsPT4 were targeted to the yeast peroxisome together with the mevalonate pathway proteins. Under galactose-induced conditions and after supplementation with OA, the production of CBGA reached 0.82 mg/L, thereby confirming that yeast organelles can act as synthetic biology devices [82].

In addition to the scientific papers indicated above, which represent some of the research carried out in this field, several patents have been published dealing with the biosynthesis of cannabinoids in yeast. Some of these patents are described below. In a patent application filed in 2016 [83], transgenic yeasts (*S. cerevisiae* and *Kluyvermyces marxianus*) capable of reconstituting the CBDA metabolic pathway of *C. sativa* were generated. This biosynthetic process was claimed to be sustainable and more environmentally friendly than synthetic production. To circumvent the low production of the precursor GPP in the yeasts, an improved mutant prenyltransferase Erg20 (K179E) that shifted the GPP:FPP ratio towards the first compound (70:30) was used in both yeasts. In *S. cerevisiae*, CBDA was produced upon the addition of hexanoic acid. In contrast, *K. marxianus* was engineered to produce hexanoyl-CoA from glucose, since the genes encoding the enzymes required for this route were introduced in this microorganism in addition to those involved in CBDA biosynthesis. In another patent, *S. cerevisiae* was engineered to incorporate the cannabinoid biosynthetic pathway, including mutant variants of Erg20. Depending on the feedstock used (OA or hexanoic acid), the Δ^9^-THCA titers were 84 mg/L and 23 mg/L, respectively. An improvement in CBGA production was achieved due to the greater availability of hexanoic acid when the endogenous FOX1 gene, which controls the β-oxidation of long-chain fatty acids, was disrupted in this yeast strain [84]. In order to produce water-soluble cannabinoids, thereby improving traditional cannabinoid production, other inventors generated a yeast expression system based on the methylotrophic yeast *K. phaffii*, which was engineered to express glycosyltransfereases from different organisms. The intracellular expression of NtGT4 (UGT 73-like glycosyltransferase from *Nicotiana tabacum*) led to the highest level of glycosylation of CBDA after the addition of 27 μM CBDA [85]. In another patent application, *S. cerevisiae* was genetically modified for the production of phytocannabinoids and phytocannabinoid analogs using a type I fatty acid synthase/PKS from the amoeba *Dictyostelium discoideum* (DiPKS). In this enzyme, which synthesizes 1-methylolivetol directly from malonyl-CoA, thereby representing an advantage regarding hexanoyl-CoA supply, point mutations were included to diminish the methylation of olivetol (G1516R) or to allow the biosynthesis of both methyl-olivetol and olivetol (G1516D/G1518A). Due to genetic engineering, the yeast strains produced 66.3 mg/L methyl-CBG, 74.26 mg/L olivetol, and 42.44 mg/L methyl-olivetol [86]. In another patent, the mevalonate pathway was overexpressed in *S. cerevisiae*. In addition, an ERG20 F96W/N127W mutant was added to provide a GPP precursor source in the cell, and pyruvate decarboxylase (PDC) from *Zymomonas mobilis* was chromosomally integrated to increase the flux from pyruvate towards acetyl-CoA. The inclusion of CsPT4 led to the production of 215.6 mg/L CBGA after feeding 1 mM OA [87]. Synthetic biology approaches in *S. cerevisiae* have also been reported in other patents [88,89], and a cell-free system for the production of cannabinoids has been patented [90].

*Candida viswanathii* was also tested as a host microorganism for the production of cannabinoids from fatty acids. The strains generated in this patent contained an engineered β-oxidation pathway and an engineered pathway for the production of CBGA, yielding 0.67 mg/L (supernatants) or 1.51 mg/L (lysates) CBGA and 5.86 mg/L (supernatants) or 12.23 mg/L (lysates) OA from oleic acid [91]. The oleaginous yeast *Y. lipolytica* has also been genetically engineered to produce the cannabinoid precursor OA. The co-expression of different enzymes, such as acetyl-CoA carboxylase, pyruvate dehydrogenase bypass, and the malic enzyme, as well as the activation of the peroxisomal β-oxidation pathway and the ATP export pathway, allowed the production of 9.18 mg/L OA in a shake-flask culture [92], which makes this yeast a promising microbial host cell for the biosynthesis of cannabinoids.

### 4.2. Biosynthesis of Cannabinoids in Filamentous Fungi: The Paradigm of P. chrysogenum

To date, there have been no reports on the biosynthetic production of cannabinoids in filamentous fungi. To the best of our knowledge, there is only one report describing the biosynthesis of OA in *Aspergillus nidulans* using fungal tandem polyketide synthases without the requirement of CsOLS and CsOAC [93], which suggested that new synthetic biology strategies can be implemented for the biosynthesis of microbial cannabinoids in filamentous fungi.

The successful trajectory of *P. chrysogenum* as industrial producer of penicillin, and the large titers achieved by this microorganism under industrial conditions [94], together with the availability of different molecular and omics tools for the genetic engineering of this filamentous fungus [54,55,57], make it very attractive as cell factory for secondary metabolite production. Based on this, we decided to use *P. chrysogenum* as a platform for cannabinoid biosynthesis and production. Therefore, we aimed to test the feasibility of filamentous fungi as cannabinoid producers by developing a *P. chrysogenum*-based biotechnological platform that allows the production of cannabinoids in submerged cultures. In order to divert the high metabolic flux resulting from penicillin biosynthesis through the heterologous cannabinoid biosynthetic pathway, a strain lacking a region that includes the penicillin biosynthetic gene cluster (*pcbAB*, *pcbC,* and *penDE* genes) and several ORFs flanking this cluster, was chosen as the parental strain. In addition, to facilitate selection upon transformation, a uridine auxotroph of this strain was used (*P. chrysogenum* Wis54-1255 npe10 pyrG−(Δpen)) [95,96]. Due to the availability of commercial CBGA and OA, we proposed a synthetic biology strategy consisting of the gradual introduction of heterologous genes required for Δ^9^-THCA biosynthesis, starting with the late biosynthetic gene THCAS (Figure 3a).

The sequence of the gene encoding the THCAS of *C. sativa* was optimized from the point of view of fungal codon usage. Previous studies executed *in silico* with the SignalP-5.0 program [97] indicated that the first 28 amino acids of this enzyme constitute a signal peptide. This signal peptide was replaced with the 19 amino acid signal peptide of the *P. chrysogenum* Pc21g02370 (with a strong similarity to aspergillopepsin apnS from *Aspergillus phoenicis*) [98]. Therefore, the cDNA of THCAS was designed to include 57 nucleotides encoding the above-mentioned signal peptide and was subcloned into the pBluescript II SK plasmid under the control of the *A. awamori gdhA* (glutamate dehydrogenase) gene promoter and the transcriptional terminator of the *S. cerevisiae cyc1* gene. This construct also included the PyrG expression cassette to allow selection upon transformation and was named pSK-F-THCAS (Figure 3a). The transformants were tested by PCR to confirm the integration of the THCAS expression cassette (not shown), and genetically modified *P. chrysogenum* transformants were named F-THCAS. These transformants were grown in PMMY medium [99] for 48 h at 26 °C and 250 rpm and supplemented with 0.1 mM CBGA. After 48 h of additional growth, mycelia were collected, broken down in liquid nitrogen to a fine powder, and resuspended in methanol. HPLC analysis was carried out following the protocol previously described in [100] with some modifications. Chromatographic separations were achieved using a Core-Shell Kinetex^®^ EVO C18 (Phenomenex, Torrance, CA, USA) analytical column (3 μm, 150 mm × 4.6 mm i.d.). The mobile phase consisted of a gradient of (methanol: acetonitrile 75:25)/water, containing 0.1% formic acid. The flow rate was set to 0.5 mL/min, and the injection volume was 20 μL. All experiments were carried out at 50 °C. Full spectra were recorded in the range of 200–400 nm. As a result of the bioconversion of CBGA, Δ^9^-THCA was detected intracellularly (Figure 3b). The authenticity of Δ^9^-THCA was confirmed by mass spectrometry using a Triple Quad 6500 System (AB Sciex LLC, Framingham, MA, USA).

In a second step, we evaluated the ability of the F-THCAS strain to carry out the prenylation of OA and biosynthesize Δ^9^-THCA (Figure 4a). To achieve this goal, we codon-optimized and overexpressed in these strains a mutant variant (Y288A/G286S) of the soluble aromatic prenyltransferase from *Streptomyces* sp. CL190 strain (StNphB), which has been reported to show improved enzyme kinetics [101], together with a codon-optimized mutant version of Erg20 (F69W/N127W), the latter leading to GPP overproduction [23].

The cDNAs encoding these two proteins were included in the pBluescript II SK plasmid under the control of the bidirectional *pcbAB-pcbC* gene promoter (controlling the expression of the *pcbAB* and *pcbC* genes of the penicillin biosynthetic gene cluster and absent in the *P. chrysogenum* Wis54-1255 npe10 pyrG−strain) and the transcriptional terminator of the *pyrG* gene. For the selection of transformants, this construct also included the phleomycin resistance cassette, thus giving rise to pSK-F-erg20-nphB (Figure 4a).

The transformants were tested by PCR to confirm the integration of the DNA fragment including the *StNphB^Y288A/G286S^* and *Erg20^F69W/N127W^* genes and the bidirectional *pcbAB-pcbC* promoter (not shown). The genetically modified *P. chrysogenum* transformants obtained in this step were named F-THCAS-erg20-nphB and were grown in PMMY medium. After 80 h of growth at 26 °C and 250 rpm, the mycelia were collected and broken down in liquid nitrogen to a fine powder to assess the formation of Δ^9^-THCA in the lysates in the presence of 1 mM OA and 0.1 mM GPP in a reaction carried out as previously described [76]. As a result of the bioconversion of OA into CBGA and the latter into Δ^9^-THCA, this cannabinoid was detected by HPLC analysis (Figure 4b) following a protocol similar to that indicated above with a different gradient. The authenticity of Δ^9^-THCA was confirmed by mass spectrometry using a Triple Quad 6500 System (SCIEX).

These preliminary results confirm that *P. chrysogenum* can biosynthesize cannabinoids using the codon-optimized plant biosynthetic genes, thus demonstrating the feasibility of filamentous fungi as cannabinoid producers. Due to the strong background of *P. chrysogenum* in the pharmaceutical industry as penicillin producer, the genetic engineering of this Ascomycete with the rest of the cannabinoid biosynthetic genes could give rise to an interesting platform for the industrial production of these compounds, although several optimization steps are still required to make it attractive for this purpose.

## 5. Conclusions

The biosynthetic production of cannabinoids in fungal hosts presents great advantages in terms of the cost and productivity of therapeutic cannabinoids. This strategy could yield greater amounts of pure products with increased stability and robustness and at a lower cost in terms of the activities and resources required compared to those necessary for the traditional process of extraction from the plant or chemical synthesis. Yeasts have emerged as suitable platforms for the industrial production of cannabinoids, and the optimization of the production process can lead to increased titers. Based upon the experience of previous industrial processes for the production of secondary metabolites, filamentous fungi could also represent an alternative host for the cannabinoid industry. Since *P. chrysogenum* has demonstrated its robustness under extreme industrial conditions in the production of penicillin, it could constitute a good candidate for the biosynthesis of other secondary metabolites. Herein, we provided evidence that this filamentous fungus could be genetically modified to produce small amounts Δ^9^-THCA after feeding cultures with CBGA or after the addition of OA to cell lysates, thereby confirming the feasibility of using this type of fungal host for this purpose. Although there is still much to be achieved in this field, including the optimization of expression cassettes, biosynthetic enzymes, and culture conditions and the redirection of metabolic fluxes, we believe that *P. chrysogenum* could become a suitable biotechnological fungal platform for the production of cannabinoids in the near future.

## Figures and Tables

**Figure 1 jof-09-00234-f001:**
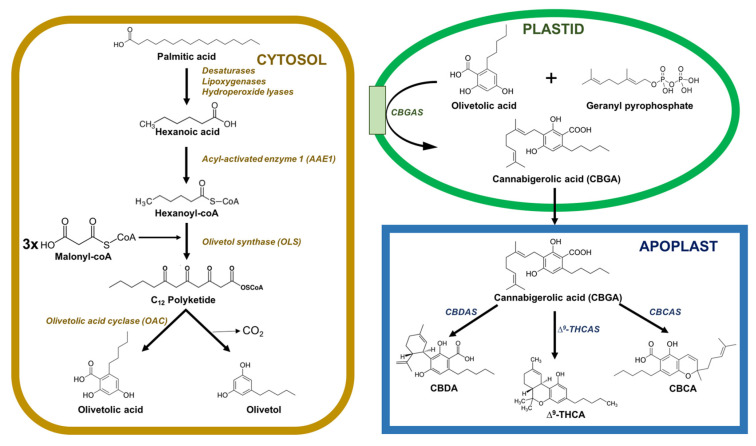
Subcellular distribution of enzymes catalyzing phytocannabinoid biosynthesis in *C. sativa*. Enzymes are located on the cytosol (yellow), plastids (green), or in the apoplastic space (blue). Abbreviations: CBCA, cannabichromenic acid; CBCAS, cannabichromenic acid synthase; CBDA, cannabidiolic acid; CBDAS, cannabidiolic acid synthase; CBGAS, cannabigerolic acid synthase; Δ^9^-THCA, Δ^9^-tetrahydrocannabinolic acid; Δ^9^-THCAS, Δ^9^-tetrahydrocannabinolic acid synthase.

**Figure 2 jof-09-00234-f002:**
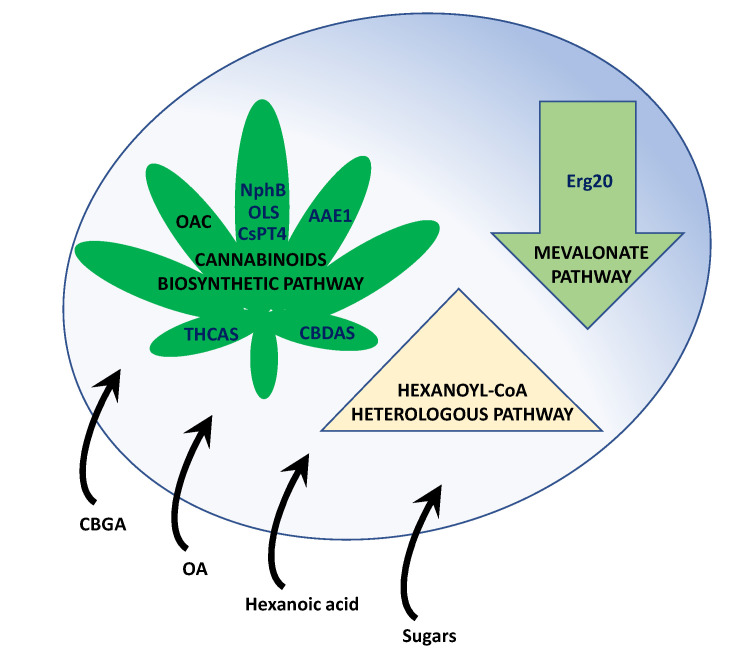
Main pathways engineered in yeasts in order to produce cannabinoids from different precursors. Abbreviations: AAE1, acyl activating enzyme; CBDAS, cannabidiolic acid synthase; CBGA, cannabigerolic acid; CBGAS, CBGA synthase; CsPT4, *C. sativa* prenyltransferase 4; Erg20, GPP synthase; NphB, aromatic prenyltransferase from *Streptomyces* sp. CL190 strain; OA, olivetolic acid; OAC, OA cyclase; OLS, olivetol synthase; THCAS, Δ^9^-tetrahydrocannabinolic acid synthase.

**Figure 3 jof-09-00234-f003:**
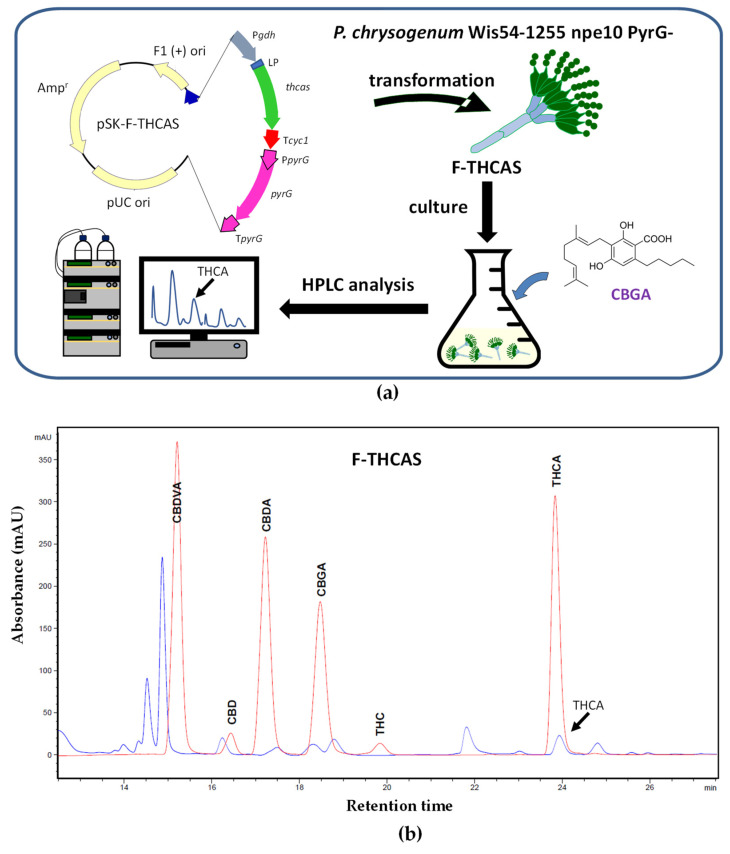
Production of Δ^9^-THCA in *P. chrysogenum* after feeding CBGA to the culture medium. (**a**) Schematic representation of the strategy followed to produce Δ^9^-THCA from CBGA in *P. chrysogenum*. Plasmid pSK-F-THCAS included: P*gdhA* (*A. awamori* glutamate dehydrogenase gene promoter), T*cyc1* (transcriptional terminator of the *S. cerevisiae cyc1* gene), LP (signal peptide), *thcas* (Δ^9^-tetrahydrocannabinolic acid synthase gene), and PyrG expression cassette. (**b**) Representative chromatogram at 270 nm of one of the transformants of the F-THCAS strain showing the formation of Δ^9^-THCA (blue line). Cannabinoid standards are included (red line).

**Figure 4 jof-09-00234-f004:**
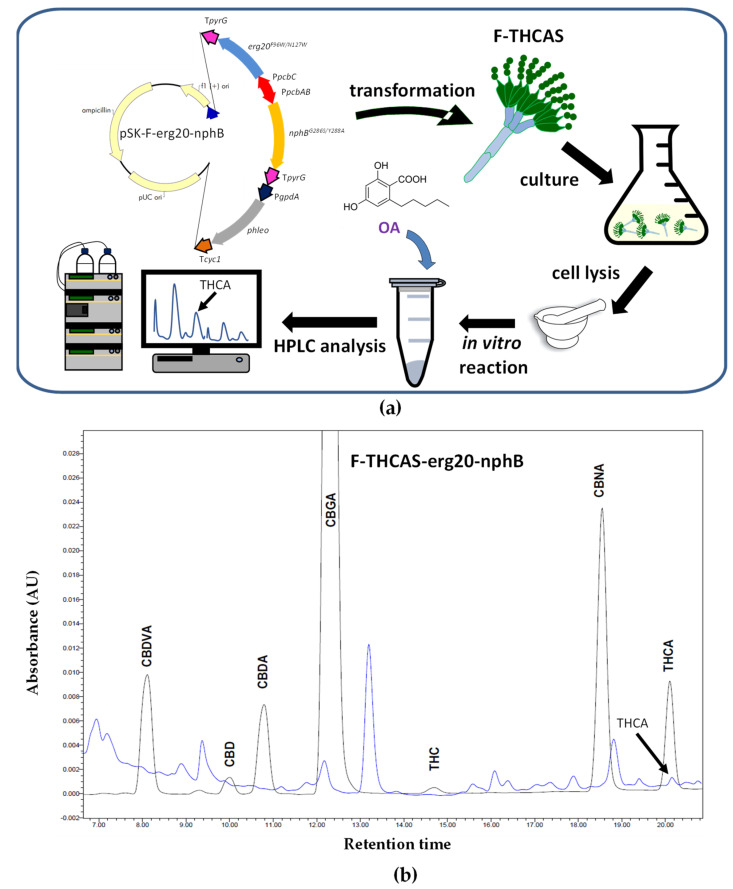
Production of Δ^9^-THCA in *P. chrysogenum* lysates after the addition of OA. (**a**) Schematic representation of the strategy followed to produce Δ^9^-THCA in *P. chrysogenum* F-THCAS. Plasmid pSK-F-erg20-nphB included: P*pcbAB*-P*pcbC* (*P. chrysogenum pcbAB-pcbC* intergenic region), T*pyrG* (transcriptional terminator of the *pyrG* gene), *erg20^F69W/N127W^* (mutant variant of the GPP synthase gene), P*gpdA* (promoter of glyceraldehyde-3-phosphate dehydrogenase (gpd) gene from *A. nidulans*), cyc1 (transcriptional terminator of the *S. cerevisiae cyc1* gene), *nphB^Y288A/G286S^* (mutant variant of the soluble aromatic prenyltransferase StNphB), and phleo (phleomycin resistance gene). (**b**) Chromatogram at 270 nm of one of the transformants of the F-THCAS-erg20-nphB strain showing the formation of Δ^9^-THCA (blue line). Chromatogram peaks corresponding to standards of cannabinoids are included (black line).

## Data Availability

Data are available on request.

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
