# Peer review of "Biotechnological Fungal Platforms for the Production of Biosynthetic Cannabinoids"

_jof, 2023, doi:10.3390/jof9020234_

Round 1

Reviewer 1 Report

The article comprehensively reviews the progress of heterogeneous production of cannabinoids in yeast, and the author even searches for relevant patents, which is more attentive. Finally, the author introduces his relevant progress in the synthesis of cannabinoids produced by filamentous fungi Penicillium Chrysogenum, proving that Penicillium Chrysogenum is also a potential cannabinoid production chassis.

However, there are still the following problems, which need to be modified or adjusted, mainly including the layout of relevant content to fit the theme, logical structure and grammatical errors in the sentence, minor writing errors, reasonable citation of literature and incorrect citation format, etc., see the notes for details.

Minor revision is recommended.

1.     In the preface, the advantages of each heterologous host (including yeast and filamentous fungi) are introduced, and compared with E. coli, the overall length is too long in part 4, and the focus can be briefly introduced.

2.     Cell free systems can be juxtaposed with yeasts and filamentous fungi in a separate section to introduce the latest developments. Or if you want to fit the title, delete it.

3.     Since Penicillium Chrysogenum can, other filamentous fungi must also be, so you can look forward to more, especially Aspergillus nidulans heterogeneous production OA.

Additional comments:

1.     Line 31-33: The sentence is too long, and there is a problem with the grammar, it is recommended to rephrase the sentence

2.     Line 44: includes(not include)

3.     Line 60: regulate(not regulates)

4.     Line 71: biosynthetic(not biosynthesis)

5.     Line 125: exert(not exerts)

6.     Line 251: The parentheses should be removed

7.     Line 252-255: The sentence is too long, the grammatical logic is not clear, it is recommended to rephrase the sentence

8.     Line 275: incorrect format of the citation

9.     Line 333 and 338: please add references reasonably

10.   Line 356: “in” should be removed

11.   Line 370:Incorrect citation in document formatting

12.   Line427:“0,1%”the comma should be replaced by a dot

13.   Line 569-571: The person's name should be put first

14.   Line 585: “A.M..” a dot should be removed

15.   Line 736 and 739: The dash after the person's name should be deleted

Author Response

Response to Reviewer 1 Comments

The article comprehensively reviews the progress of heterogeneous production of cannabinoids in yeast, and the author even searches for relevant patents, which is more attentive. Finally, the author introduces his relevant progress in the synthesis of cannabinoids produced by filamentous fungi Penicillium Chrysogenum, proving that Penicillium Chrysogenum is also a potential cannabinoid production chassis.

However, there are still the following problems, which need to be modified or adjusted, mainly including the layout of relevant content to fit the theme, logical structure and grammatical errors in the sentence, minor writing errors, reasonable citation of literature and incorrect citation format, etc., see the notes for details.

Minor revision is recommended.

Point 1: 1.     In the preface, the advantages of each heterologous host (including yeast and filamentous fungi) are introduced, and compared with E. coli, the overall length is too long in part 4, and the focus can be briefly introduced.

Response 1: Thank you for the comment. We have reduced the length of this section as suggested.

Point 2: 2.     Cell free systems can be juxtaposed with yeasts and filamentous fungi in a separate section to introduce the latest developments. Or if you want to fit the title, delete it.

Response 2: Thank you for the comment. In order to fit the title we have removed the paragraphs regarding the production of cannabinoids from yest cell free systems.

Point 3: 3.     Since Penicillium Chrysogenum can, other filamentous fungi must also be, so you can look forward to more, especially Aspergillus nidulans heterogeneous production OA.

Response 3: Thank you for the comment. We have included a paragraph about the heterologous production of OA in A. nidulans at the beginning of section 4.2.

“To date, there are no reports about the biosynthetic production of cannabinoids in filamentous fungi. To the best of our knowledge, there is only one report describing the biosynthesis of OA in Aspergillus nidulans using fungal tandem polyketide synthases without the requirement of CsOLS and CsOAC (Okorafor et al., 2021), which suggested that new synthetic biology strategies can be implemented for the biosynthesis of microbial cannabinoids in filamentous fungi”.

Additional comments:

Point 1.     Line 31-33: The sentence is too long, and there is a problem with the grammar, it is recommended to rephrase the sentence.

Response 1. We have rephrased the sentence as suggested.

Point 2.     Line 44: includes(not include)

Response 2. This typo has been corrected.

Point 3.     Line 60: regulate(not regulates)

Response 3. This typo has been corrected.

Point 4.     Line 71: biosynthetic(not biosynthesis)

Response 4. This typo has been corrected.

Point 5.     Line 125: exert(not exerts)

Response 5. This typo has been corrected.

Point 6.     Line 251: The parentheses should be removed

Response 6. The parenthesis has been removed.

Point 7.     Line 252-255: The sentence is too long, the grammatical logic is not clear, it is recommtended to rephrase the sentence

Response 7. This sentence has been removed, since it was related to cell free systems (see Point 2 in the main comments).

Point 8.     Line 275: incorrect format of the citation

Response 8. This sentence has been removed, since it was related to cell free systems (see Point 2 in the main comments).

Point 9.     Line 333 and 338: please add references reasonably

Response 9. A reference has been included regarding yeast peroxisomes: Zhang Q, Zeng W, Xu S, Zhou J. Metabolism and strategies for enhanced supply of acetyl-CoA in Saccharomyces cerevisiae. Bioresour Technol. 2021 Dec;342:125978. doi: 10.1016/j.biortech.2021.125978. Epub 2021 Sep 20. PMID: 34598073.

Point 10.   Line 356: “in” should be removed

Response 10. The sentence has been rephrased.

Point 11.   Line 370:Incorrect citation in document formatting

Response 11. This mistake has been corrected.

Point 12.   Line427:“0,1%”the comma should be replaced by a dot

Response 12. This mistake has been corrected.

Point 13.   Line 569-571: The person's name should be put first

Response 13. This mistake has been corrected.

Point 14.   Line 585: “A.M..” a dot should be removed

Response 14. This mistake has been corrected.

Point 15.   Line 736 and 739: The dash after the person's name should be deleted

Response 15. This mistake has been corrected.

Reviewer 2 Report

The Review by Garcia-Estrada et al. concerns the biosynthesis of cannabinoids, including various approaches, their application, biological activity etc. The special attention is paid to the application of modified fungi for production of cannabinoids. This subject may be regarded as novel, actual; it would be interesting to a wide readership, to researchers working in mycology, organic chemistry, biology, medicine and other branches of Science. The text is well organized and illustrated; it was written by simple and understandable language. The literature included sources of 1974-2022 years. In general, this Manuscript makes a favorable impression. I have only several Notes (see below), which may be easily fixed by the Authors, so I would like to recommend a Minor Revision before submission.

Notes to the Authors:

1) Line 54: “…para-positioned”. In relation to what?

2)      L.54: “…aralkyl” ??? Arylalkyl?

3)      In the text use “.DELTA.9” with uppercase number.

4)      Several typos: ll. 68, 171, 381, 421, 428, 470.

5)      References [83-90]: for patents use number, country, company.

Author Response

Response to Reviewer 2 Comments

The Review by Garcia-Estrada et al. concerns the biosynthesis of cannabinoids, including various approaches, their application, biological activity etc. The special attention is paid to the application of modified fungi for production of cannabinoids. This subject may be regarded as novel, actual; it would be interesting to a wide readership, to researchers working in mycology, organic chemistry, biology, medicine and other branches of Science. The text is well organized and illustrated; it was written by simple and understandable language. The literature included sources of 1974-2022 years. In general, this Manuscript makes a favorable impression. I have only several Notes (see below), which may be easily fixed by the Authors, so I would like to recommend a Minor Revision before submission.

Notes to the Authors:

Point 1: 1) Line 54: “…para-positioned”. In relation to what?

Response 1: The isoprenyl, alkyl, or aralkyl side chain are in para position in relation to the resorcinol core.

Point 2: 2)      L.54: “…aralkyl” ??? Arylalkyl?

Response 2: Aralkyl is a term that refers to an alkyl that is substituted with an aryl group.

Point 3: 3)      In the text use “.DELTA.9” with uppercase number.

Response 3: Thank you for the comment. We have corrected it along the manuscript.

Point 4: 4)      Several typos: ll. 68, 171, 381, 421, 428, 470.

Response 4: We have corrected typos along the manuscript.

Point 5: 5)      References [83-90]: for patents use number, country, company.

Response 5: Thank you for the comment. We have modified those references as suggested.
